# Quercetin Prevents LPS-Induced Oxidative Stress and Inflammation by Modulating NOX2/ROS/NF-kB in Lung Epithelial Cells

**DOI:** 10.3390/molecules26226949

**Published:** 2021-11-17

**Authors:** Ok-Joo Sul, Seung Won Ra

**Affiliations:** 1Biomedical Research Center, University of Ulsan, Ulsan 44030, Korea; suloj@ulsan.ac.kr; 2Department of Internal Medicine, Ulsan University Hospital, University of Ulsan College of Medicine, Ulsan 44030, Korea

**Keywords:** oxidative stress, inflammation, quercetin, NADPH oxidases, reactive oxygen species

## Abstract

Oxidative stress caused by the production of reactive oxygen species (ROS) plays a major role in inflammatory processes. We hypothesized that modulation of ROS via quercetin may protect against oxidative stress and inflammation. Thus, this study aimed to investigate the effects of quercetin on oxidative stress and inflammation in lung epithelial A549 cells. The lipopolysaccharide (LPS)-induced elevation of intracellular ROS levels was reduced after quercetin treatment, which also almost completely abolished the mRNA and protein expression of nicotinamide adenine dinucleotide phosphate oxidase 2 (NOX2) induced by LPS stimulation. In addition, quercetin suppressed the nuclear translocation of nuclear factor kappa B (NF-κB) and reduced levels of inflammatory cytokine tumor necrosis factor (TNF)-α, interleukin (IL)-1, and IL-6, which had increased significantly after LPS exposure. Our data demonstrated that quercetin decreased ROS-induced oxidative stress and inflammation by suppressing NOX2 production.

## 1. Introduction

Acute lung injury (ALI) leading to acute respiratory distress syndrome (ARDS) is a severe clinical disease that involves widespread inflammation in the lung tissues and subsequent development of lung dysfunction [1]. ALI is characterized by disruption of the alveolar–capillary interface, neutrophil recruitment to the lung, and release of chemokines and proinflammatory cytokines by immune cells and non-immune cells such as alveolar epithelial cells [2,3].

Inflammation is a host immune response to chemical and physical injury or infection, usually caused by various bacteria. However, excessive inflammation can cause uncontrolled production of inflammatory mediators such as cytokines, chemokines, and reactive oxygen species (ROS) and plays a key role in the development of ALI [4].

Excess production of ROS relative to antioxidant defense causes oxidative stress. Several studies have shown that ROS-generating nicotinamide adenine dinucleotide phosphate (NADPH) oxidases (NOXs) are involved in lipopolysaccharide (LPS)-induced lung inflammation; among members of the large NOX family, NOX2 is the major source for inflammation-associated ROS production in lung inflammation [5,6,7].

Many studies have shown that ROS production leading to oxidative stress plays a major role in inflammatory processes [8,9]. Oxidative stress can activate a variety of transcription factors which lead to the differential expression of some genes involved in inflammatory pathways. Nuclear factor kappa B (NF-κB), a major redox-sensitive transcription factor, exists in an inactive form through its association with the IκB proteins in the cytoplasm. ROS drive modifications of IκB proteins; e.g., phosphorylation by the inhibitor of NF-κB kinase (IKK) leads to its degradation. These events allow active NF-κB to translocate to the nucleus, where it induces the expression of several molecules, such as tumor necrosis factor (TNF)-α, interleukin (IL)-1β, and IL-6, which are involved in the inflammatory process of ALI [10,11].

Flavonoids have been reported to serve as potent antioxidants that can reduce the risk of many chronic diseases [12]. Quercetin, 3,3′,4′,5,7-pentahydroxyflavone with three rings and five hydroxyl groups (Figure 1A), is a dietary bioflavonoid that is abundant in fruits and vegetables and shows significant antioxidant [13,14], antiviral [15], anti-carcinogenic [16], and anti-inflammatory activities [17]. In fact, because of its ability to reduce oxidative stress quercetin has been reported to show a protective effect against LPS-induced lung injury. Quercetin has been shown to decrease the production of the LPS-stimulated inflammatory cytokines TNF-α, IL-1β, and IL-6 in RAW264.7 macrophages [18,19] and to modulate several inflammatory genes regulated by NF-κB [20]. It has also been shown to significantly reduce NOX-derived oxidative stress in LPS-stimulated macrophages [21]. However, the precise mechanisms by which quercetin prevents oxidative stress and inflammation in LPS-induced lung epithelial cells remain unclear. In this study, we sought to evaluate the anti-inflammatory and antioxidant effects of quercetin on LPS-induced lung epithelial cells and to elucidate the potential underlying mechanisms.

## 2. Materials and Methods

### 2.1. Reagents and Antibodies

Quercetin, LPS, and *N*-acetyl-l-cysteine (NAC), diphenyleneiodonium (DPI), and 3-(4,5-dimethylthiazol-2-yl)-2,5-diphenyltetrazolium bromide (MTT) were purchased from Sigma-Aldrich (St. Louis, MO, USA). The NE-PER Nuclear and Cytoplasmic Extraction Reagent Kit and 2′,7′-dichlorofluorescein diacetate (H2DCFDA) were obtained from Thermo Fisher Scientific (Waltham, MA, USA). The enzyme-linked immunosorbent assay (ELISA) kits for determination of TNF-α and IL-6 were obtained from R&D systems Inc. (Minneapolis, MN, USA). Antibodies against NOX2 and β-actin were purchased from Enzo Life Sciences (Farmingdale, NY, USA) and Sigma Chemical, respectively. Antibodies against NF-κB, p65, IκB, phospho-IκB and Lamin B1 were obtained from Cell Signaling (Danvers, MA, USA).

### 2.2. Cell Culture

Human alveolar epithelial A549 cells (KCLB, Seoul, Korea) were grown in RPMI 1640 medium (Gibco, Grand Island, NY, USA) supplemented with 10% FBS and 1% penicillin-streptomycin solution at 37 °C in a humidified atmosphere containing of 5% CO_2_. A549 cells were pretreated with quercetin (10 μM) for 4 h and then stimulated with or without LPS (10 µg/mL) for an additional 6 h. 

### 2.3. Cell Viability Assay

Cell viability was determined by MTT assay. A549 cells were cultured in a 96-well plate at 3 × 10^4^ cells per well and grown to 80% confluency. The cells were then pretreated with quercetin at various concentrations (0, 10, 20, 50, and 100 μM) for 4 h and stimulated with or without LPS at various concentrations (0, 10, 20, 30, and 50 μM). The cells were incubated with 0.5 mg/mL of thiazolyl blue tetrazolium bromide (MTT, Sigma-Aldrich) for 3 h. After removing the MTT and adding 100 μL of dimethylformamide (DMSO), the optical density of the samples was read at 540 nm on a spectrophotometer.

### 2.4. RNA Isolation and Quantitative Polymerase Chain Reaction

Total RNA was extracted from A549 cells using QIAzol reagent, then 1 μg of total RNA was reverse-transcribed into cDNA using oligo (dT) primers and M-MLV reverse transcriptase. The synthesized cDNA was amplified with SYBR Green qPCR Master Mix (Enzynomics, Daejeon) to perform real-time quantitative PCR (RT-qPCR) using an ABI 7500 Fast Real-Time PCR System (Applied Biosystems, Carlsbad, CA, USA). The primers used in qPCR are listed in Table 1.

### 2.5. Determination of Intracellular ROS

A549 cells were pretreated with quercetin (10 μM) for 6 h and then stimulated with or without LPS (10 µg/mL) for an additional 4 h. The cultured medium was then replaced with phosphate-buffered saline (PBS) containing 5 μM CM-H2DCFDA (C6827; Invitrogen, Carlsbad, CA, USA) for 30 min at 37 °C. Fluorescence was detected by flow cytometry (FACSCanto II), and the data were analyzed by FlowJo V10 software (Tree Star Inc., San Carlos, CA, USA). To measure ROS using confocal microscopy, A549 cells were grown on glass cover slips in 24-well plates. The cells were pretreated with quercetin, stimulated in the absence or presence of LPS, and stained with 10 μM CMH2DCFDA for 15 min. The cover slips were removed from the wells of the plate and placed on a slide for analysis. Fluorescence images were analyzed using an Olympus FV1200 confocal microscope (Olympus, Tokyo, Japan). 

### 2.6. Nuclear and Cytosolic Fractionation

Cells were fractionated using a nuclear and cytosol fractionation kit for detecting NF-κB p65 according to the manufacturer’s protocol. Proteins were separated by SDS-PAGE and transferred to nitrocellulose membranes. The membranes were blocked with 5% skim milk in Tris-buffered saline containing 0.1% Tween 20% (1× TBS-T) for 1 h at room temperature, followed by overnight incubation at 4 °C with primary antibodies against NF-κB p65 and Lamin B1. After washing with 1× TBS-T, the membranes were incubated with HRP-conjugated secondary antibodies for 1 h at room temperature and developed using the ECL detection system (GE Healthcare, Waukesha, WI, USA).

### 2.7. Western Blot Analysis

The cells were lysed with lysis buffer (50 mM Tris-HCl, pH 8.0; 150 mM NaCl; 1 mM EDTA; 0.5% Nonidet P-40; 0.01% protease inhibitor mixture, 0.5 mM phosphatase inhibitor). The lysates (20 µg) were loaded onto SDS-PAGE and transferred onto the nitrocellulose membrane. The membranes were blocked with 5% skim milk for 1 h at room temperature (RT) and incubated with polyclonal antibodies to IκBα, phospho-IκB (Santa-Cruz), and β-actin (Sigma-Aldrich). The membranes were washed with 1x TBS-T and incubated with HRP-conjugated secondary antibodies for 1 h at room temperature, followed by development with the ECL detection system (GE Healthcare, Waukesha, WI, USA). Band intensities were quantified by densitometry using the Image J program.

### 2.8. Enzyme-Linked Immunosorbent Assay 

Supernatants were collected and the production of IL-6 and TNF-*α* was measured by a sandwich ELISA obtained from R&D Systems (Minneapolis, MN, USA). ELISA was performed according to the manufacturer’s protocol.

### 2.9. Statistical Analyses

All values are expressed as means ± SEM. Student’s t-test was used to evaluate differences between samples of interest and the corresponding controls. A *p*-value of <0.05 was considered to be of statistical significance.

## 3. Results

### 3.1. Effects of Quercetin on Cell Viability

To evaluate whether quercetin reduces LPS-induced oxidative stress and inflammation, we first assessed the effect of quercetin on human lung epithelial A549 cell viability. Cells were treated with quercetin at various concentrations (0–100 μM) for 4 h before addition of LPS (0–50 μg/mL) for 6 h. Cell viability was measured by MTT assay. This assay is commonly used for assessing the viability of cells and measuring the cytotoxicity of compounds. As shown in Figure 1B, no significant change in cell viability was observed with quercetin treatment at doses of 10, 20, and 50 μM, while suppression of cell viability was detected at a dose of 100 μM. Previous work in our lab showed similar antioxidant effects at concentrations of 10 and 20 µM. Thus, we used quercetin at a concentration of 10 μM for the subsequent studies. In addition, LPS had no effect on ROS generation at concentrations below 10 µg/mL in A549 cells, while ROS generation was induced by LPS at concentrations of 10–50 µg/mL. As there were no significant differences at these concentrations, it was decided to continue further experiments with 10 µg/mL LPS.

### 3.2. Quercetin Prevents LPS-Induced Oxidative Stress in Lung Epithelial Cells

Since quercetin has been reported to be a potential antioxidant, we first evaluated whether it inhibited LPS-induced oxidative stress. To investigate whether quercetin decreases oxidative stress in lung epithelial cells, we measured intracellular ROS levels in LPS-treated cells by using DCF-DA, which is oxidized by ROS to fluorescence DCF. Cells were pretreated with quercetin (10 μM) for 4 h followed by LPS (10 µg/mL) for another 6 h, then stained with DCF-DA. Intracellular ROS was then detected using confocal microscopy and flow cytometry (Figure 2A,B, respectively). Interestingly, we observed that cells pretreated with quercetin exhibited markedly decreased intracellular ROS production in response to LPS. These results show that quercetin exhibits antioxidant effects on LPS-induced oxidative stress. 

### 3.3. Quercetin Prevents LPS-Induced Oxidative Stress by Suppressing NOX2 Production

To evaluate the precise mechanisms by which quercetin prevents oxidative stress, A549 cells were pretreated with quercetin for 4 h followed by LPS stimulation for 6 h. Excess production of ROS relative to antioxidant defense can result in oxidative stress. Since one of the main sources of ROS in cells is NOX2, and NOX2 is expressed in pulmonary endothelial cells, we first assessed whether quercetin can reduce ROS production by inhibiting the NOX2 level [22,23]. As expected, both the protein and the transcript level of NOX2 significantly increased in the presence of LPS; however, pre-treatment of the cells with quercetin for 4 h followed by LPS for 6 h significantly prevented LPS-induced NOX2 generation when compared to the group treated with LPS alone (Figure 3A). Treatment with NAC (5 mM), an ROS scavenger, and DPI (10 μM), a NOX inhibitor, also abolished the intracellular ROS generation increased by LPS stimulation (Figure 3B), suggesting that quercetin inhibits the oxidative stress induced by LPS through suppression of NOX2 production.

### 3.4. Quercetin Inhibits LPS-Induced Inflammation in Lung Epithelial Cells

Proinflammatory cytokines such as TNF-α, IL-1β, and IL-6 play important roles in the occurrence and development of inflammation [24]. To evaluate the anti-inflammatory activity of quercetin on LPS-stimulated cells, we measured the levels of these proinflammatory cytokines. Western blotting and RT-qPCR analysis showed that LPS stimulation significantly elevated the protein and mRNA levels of IL-1β and IL-6, especially that of TNF-α (Figure 4A,B). However, production of LPS-induced cytokines was inhibited markedly by treatment with quercetin. Both the mRNA and protein levels of TNF-α, IL-1β and IL-6 in the LPS plus quercetin group were lower than those obtained with LPS treatment alone. Treatment with NAC (5 mM) also inhibited both mRNA and protein levels of these proinflammatory cytokines increased by LPS (Figure 4A,B). Therefore, quercetin regulates LPS-induced inflammation by reducing oxidative stress.

### 3.5. Quercetin Significantly Decreases LPS-Induced Nuclear Translocation of NF-κB in Lung Epithelial Cells

NF-κB signaling plays a pathogenic role in many inflammatory lung diseases [10,11], and NF-κB mediates the induction of inflammatory cytokines such as TNF-α, IL-1, and IL-6. To determine whether the anti-inflammatory action of quercetin occurs via NF-κB signaling, we examined the nuclear translocation of NF-κB (p65) and the degradation of its upstream signaling molecule IκBα in lung epithelial cells. As shown in Figure 5A,B, IκBα was degraded 10 min following LPS treatment, which was followed by significant movement of the NF-κB dissociated from IκBα from the cytoplasm to the nucleus. However, pretreatment with quercetin significantly prevented the LPS-induced IκBα degradation and the translocation of NF-κB to the nucleus. We observed that quercetin prevented IκBα degradation and the nuclear translocation of NF-κB (p65), indicating that quercetin can inhibit NF-κB signaling. In addition, NAC, a scavenger of ROS, similarly inhibited the nuclear translocation of NF-κB (p65). These results show that the anti-inflammatory effects of quercetin may result from inhibition of NF-κB.

## 4. Discussion

ALI is characterized by acute lung inflammation that results in disruption of lung endothelial and epithelial barriers. One of the most important determinants of lung injury severity is the extent of injury to the epithelial barrier. Therefore, we used A549 typical human alveolar epithelial cells, which have been implicated in the pathogenesis of ALI [25]. Epithelial cells (as well as macrophages and neutrophils) generate cytokines and chemokines in response to inflammatory stimuli [26,27], and the airway epithelium modulates lung inflammation and injury through NF-κB signaling [28]. Prolonged ROS production is considered to be important for the progression of many inflammatory diseases [29]. Since LPS can promote oxidative stress and inflammation, and is known to be a primary risk factor for ALI, we first determined whether LPS increases ROS production and whether quercetin reduces the LPS-induced oxidative stress in alveolar epithelial cells. Several reports have shown that quercetin protects gastric epithelial GES-1 cells from oxidative damage both in vitro and in vivo [13] and reduces paraquat-induced oxidative damage by regulating the expression of antioxidant genes in A549 cells [14]. 

As potent antioxidants, flavonoids reduce the risk of many chronic diseases [12], and it is important to investigate their preventive effects against inflammation-associated disorders. In this regard, modulation of oxidative stress via quercetin may offer protection against inflammation that leads to lung disease progression, as inflammation is triggered by oxidative stress. Consistent with these findings, we observed that quercetin reduced the levels of intracellular ROS production in response to LPS, indicating that quercetin protects against LPS-induced oxidative stress. Furthermore, because the primary ROS-generating enzyme, NOX, is a complex enzyme system that is present in phagocytes and epithelial cells, we sought to evaluate the mechanisms by which quercetin protects against LPS-induced oxidative stress. Thus, we investigated the mechanism by which quercetin ameliorated LPS-increased oxidative stress. 

NOX plays a significant role in ROS-induced lung inflammation [5]. In addition, because NADPH oxidase is the primary ROS-generating enzyme and is a complex enzyme present in phagocytes and epithelial cells, we evaluated whether NOX is associated with LPS-induced ROS production and whether quercetin can suppress the LPS-induced ROS production. Interestingly, our results showed that quercetin almost completely abolished the mRNA and protein levels of NOX2 induced by LPS stimulation in lung epithelial A549 cells, indicating that quercetin prevents LPS-induced oxidative stress by downregulation of NADPH oxidase in lung epithelial A549 cells. Thus, LPS-induced oxidative stress can result from increased oxidative stress via increased NOX2 expression. Conversely, quercetin reversed the LPS-induced oxidation by regulating NOX2 expression. 

Many studies have shown that ROS production leading to oxidative stress plays a major role in inflammatory processes [8,30]. Because inflammation is triggered by oxidative stress and is the cause of many lung diseases, we investigated whether quercetin can inhibit LPS-induced inflammation. In this study, we observed that quercetin reduced the levels of the inflammatory cytokines TNFα, IL-1, and IL-6, which were increased significantly after LPS exposure. *N*-acetylcysteine (NAC), an important cellular antioxidant, inhibits the expression of several proinflammatory genes [31]. Consistent with this, we found that NAC significantly decreased the levels of these cytokines in alveolar epithelial cells. Several studies have shown that quercetin can downregulate ROS-induced oxidative stress and inflammation through regulating other signaling pathways. Veith et al. showed that quercetin increased the antioxidant response through by increasing Nrf2 activity and reducing the levels of pro-inflammatory cytokines [32]. Quercetin also suppressed the inflammatory enzymes cyclooxygenase (COX) and lipooxygenase, which in turn decreased inflammatory mediators containing prostaglandins and leukotrienes [33,34]. In addition, quercetin inhibited NO and iNOS expression by regulating the NF-kB/HO pathway in LPS-exposed BV2 cells [35]. Xiong et al. reported that quercetin ameliorated LPS-induced inflammation in HGFs by activating PPAR-γ and suppressing the activation of NF-κB [36]. Thus, our results suggest that quercetin decreases LPS-induced inflammation, thereby exerting a protective effect against LPS-induced inflammation, at least partially by modulating oxidative stress. 

NF-κB regulates the expression of several inflammatory cytokines, such as TNF-α, IL-1β, and IL-6, which are involved in the inflammatory processes underlying ALI [11,37]. Accordingly, in this study, we further investigated the role of quercetin in LPS-induced NF-kB activation. NAC inhibits the upstream events that lead to NF-kB activation [38]. Our results revealed that, similar to the effects of NAC in abolishing LPS-induced activation of NF-κB, quercetin suppressed the nuclear translocation of NF-κB, suggesting that the anti-inflammatory effects of quercetin may induce inhibition of NF-κB.

Numerous studies have shown that quercetin has the potential to treat lung diseases due to its strong antioxidant and free radical scavenging effects; in particular, quercetin could be an effective treatment for idiopathic pulmonary fibrosis (IPF) by restoring lung redox homeostasis and preventing inflammation [32]. Quercetin could also alleviate acute lung injury by decreasing oxidative stress and enhancing the activity of antioxidant enzymes [39]. When the inhibitory effects of quercetin were compared with those of vitamin E in the context of bleomycin-induced pulmonary fibrosis in rats, the antifibrotic activity of quercetin was higher than that of vitamin E [40], a well-known antioxidant. Additionally, Bidian et al. evaluated the antioxidant effects of quercetin and curcumin on pleural inflammation induced by carrageenan and showed that in lung tissue, quercetin showed superior beneficial properties while in serum, curcumin had higher antioxidant effects compared to quercetin [41]. Many studies showed promising therapeutic potential for quercetin in treating lung disease; however, further research into quercetin is needed before pharmacological application.

In conclusion, our results reveal that quercetin inhibits oxidative stress by reducing the expression of NOX2 and that it suppresses the nuclear translocation of NF-κB, thereby inhibiting the expression of proinflammatory cytokines such as TNF-α and IL-1β (Figure 6). Thus, quercetin could be a potential therapeutic agent for lung diseases associated with inflammation and oxidative stress. 

## Figures and Tables

**Figure 1 molecules-26-06949-f001:**
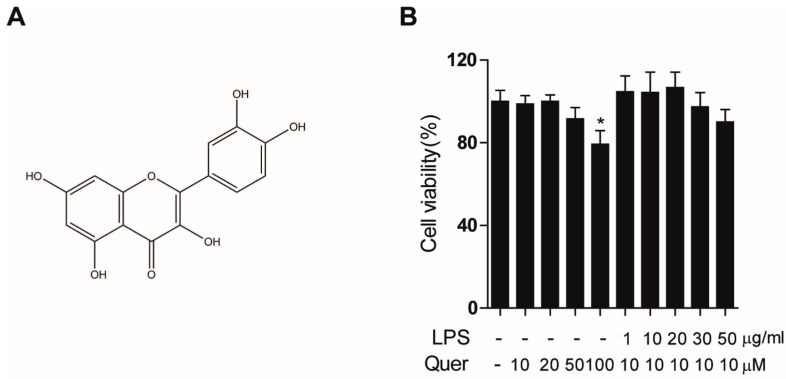
Structure of quercetin and its effect on cell viability: (**A**) Chemical structure of quercetin; (**B**) Lung epithelial A549 cells were treated with quercetin at various concentrations (0, 10, 20, 30 and 50 μM) for 4 h before addition of LPS (0, 1, 10, 20, 30 and 50 μg/mL) for 6 h. Cell viability was measured using the MTT assay. * *p* < 0.05 in comparison with non-treated cells. Similar results were obtained in three independent experiments.

**Figure 2 molecules-26-06949-f002:**
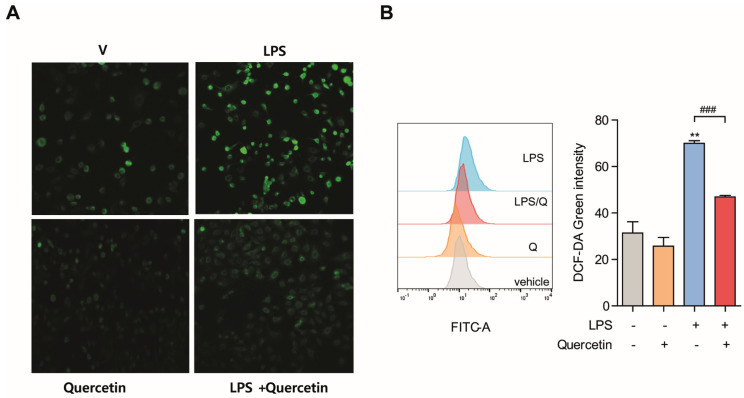
Quercetin prevents LPS−induced oxidative stress in lung epithelial cells. A549 cells were pretreated with quercetin (10 μM) for 4 h and then stimulated with LPS (10 μg/mL) for an additional 6 h. Production of intracellular ROS was determined by DCF-DA staining using confocal microscopy (**A**) and flow cytometry (**B**). Scale bar = 10 μm. ** *p* < 0.001 in comparison with non-treated cells. ^###^ *p* < 0.001 compared with LPS-treated cells. Similar results were obtained in three independent experiments.

**Figure 3 molecules-26-06949-f003:**
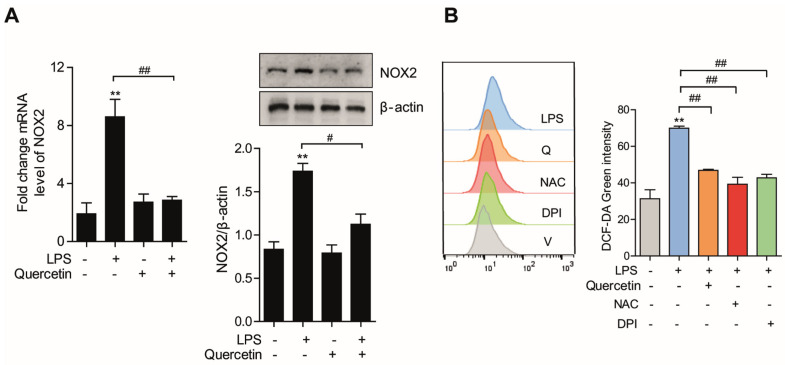
Quercetin prevents LPS−induced oxidative stress by suppressing NOX production. A549 cells were pretreated with quercetin (10 μM) for 4 h and then stimulated with LPS (10 μg/mL) for an additional 6 h. (**A**) The mRNA and protein levels of NOX2 were determined by qPCR and western blot, respectively. Band intensity was determined using the Image J program, then β-actin was used as the loading control protein to normalize target protein expression. The fold change was calculated by dividing the normalized expression from each lane by the normalized expression of the control sample. (**B**) Cells were pretreated with the NOX2 inhibitor NAC (5 mM) or DPI (10 μM) for 30 min or with quercetin (10 μM) for 4 h. Then, the cells were challenged with LPS (10 μg/mL) for another 6 h. Intracellular ROS was stained with DCF-DA and measured by flow cytometry. ** *p* < 0.01 in comparison with non-treated cells. ^#^ *p* < 0.05; ^##^ *p* < 0.01 compared with LPS-treated cells. Similar results were obtained in three independent experiments.

**Figure 4 molecules-26-06949-f004:**
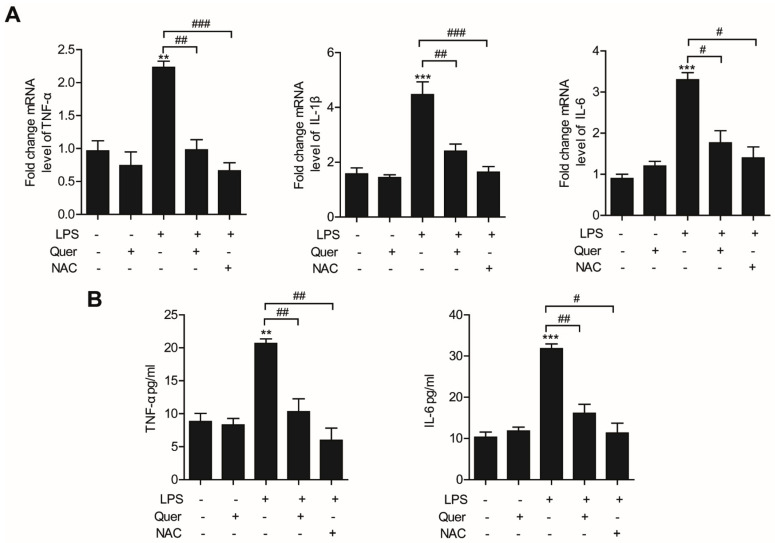
Quercetin inhibits LPS−induced inflammation in lung epithelial cells. A549 cells pretreated with quercetin (10 μM) for 4 h and then stimulated with LPS (10 μg/mL) for an additional 6 h. (**A**) mRNA levels of TNF-α, IL-1β, and IL-6 were determined by qPCR. (**B**) Cells were pretreated with the NOX2 inhibitor NAC (5 mM) for 30 min followed by incubation with quercetin (10 μM) for 4 h. Then, the cells were challenged with LPS (10 μg/mL) for another 6 h. The conditioned supernatants were harvested, and the secreted levels of TNF-α and IL-6 were assessed by ELISA. ** *p* < 0.01; *** *p* < 0.001 in comparison with non-treated cells. ^#^ *p* < 0.05; ^##^ *p* < 0.01; ^###^ *p* < 0.001 compared with LPS-treated cells. Similar results were obtained in three independent experiments.

**Figure 5 molecules-26-06949-f005:**
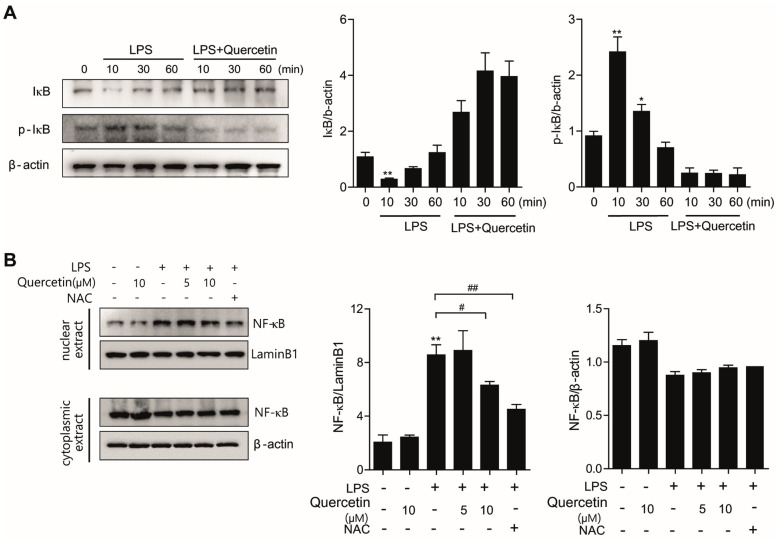
Quercetin significantly decreased LPS−induced nuclear translocation of NF-κB in lung epithelial cells. A549 cells were pretreated with quercetin (10 µM) for 4 h before treatment with 100 ng/mL LPS for 6 h. (**A**) The protein levels of IκB were determined by western blot. (**B**) Cells were pretreated with the NOX2 inhibitor NAC (5 mM) for 30 min followed by incubation with quercetin (10 μM) for 4 h. Then, the cells were challenged with LPS (10 μg/mL) for another 6 h. The protein levels of NF-κB were determined by western blotting in cytosolic or nuclear extracts. β-actin and lamin B were used as the loading control proteins to normalize the expression of cytosolic and nuclear proteins. Values are shown as means ± SEM. * *p* < 0.05; ** *p* < 0.01 in comparison with non-treated cells. ^#^ *p* < 0.05; ^##^ *p* < 0.01 compared with LPS-treated cells. Similar results were obtained in three independent experiments.

**Figure 6 molecules-26-06949-f006:**
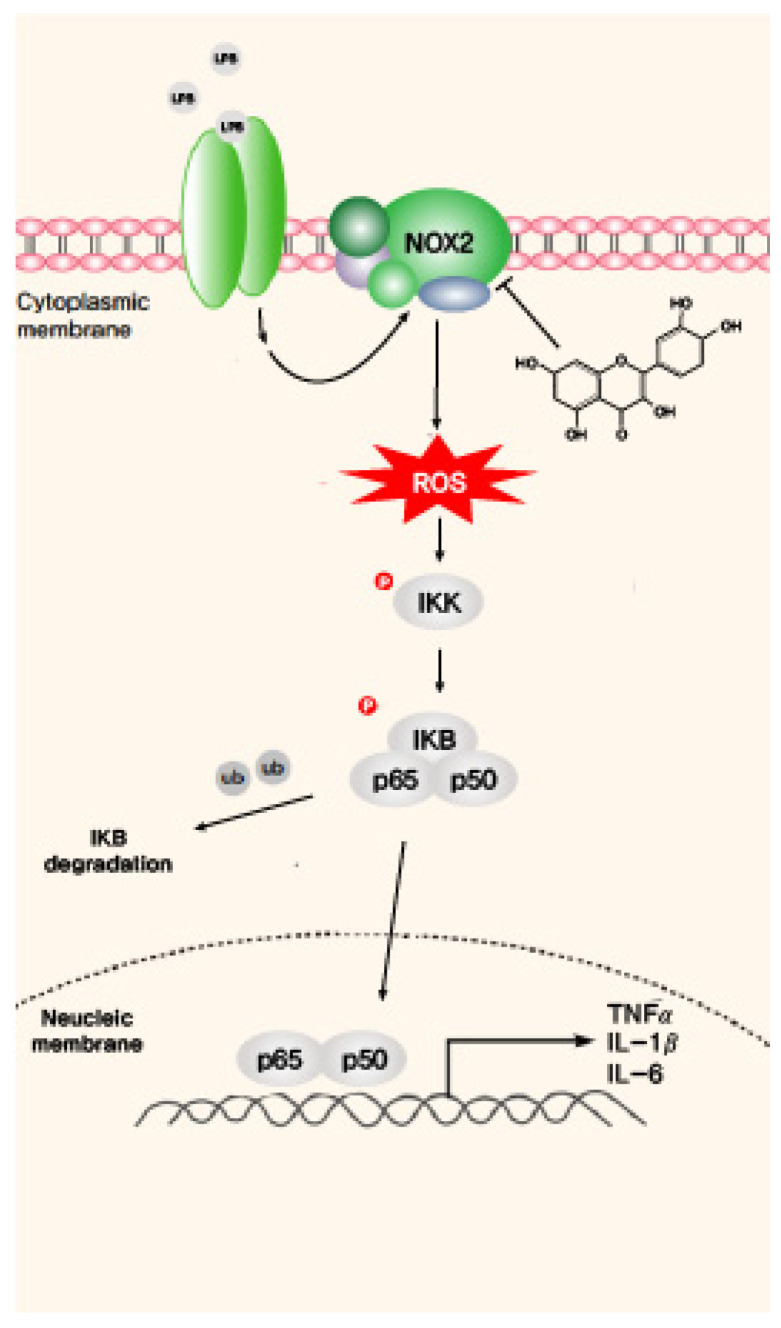
Quercetin decreases reactive oxygen species (ROS)-induced oxidative stress and inflammation by suppressing NADPH oxidase 2 (NOX2) production. Lipopolysaccharide (LPS) activates its receptor and transmits a signal to generate ROS via NOX2. The LPS-induced ROS levels enhance IκBα degradation and the translocation of NF-κB to the nucleus. NF-κB regulates the expression of several inflammatory cytokines, such as TNF-α, IL-1β, and IL-6. Quercetin prevents LPS-induced NOX2 expression and suppresses IκBα degradation and the nuclear translocation of NF-κB, resulting in a reduction in the levels of the inflammatory cytokines.

**Table 1 molecules-26-06949-t001:** The primers used in qPCR.

Gene Name	Forward Primer Sequence (5′-3′)	Reverse Primer Sequence (5′-3′)
NOX2	GCAGCCTGCCTGAATTTCA	TGAGCAGCACGCACTGGA
TNFα	CTCTCTCTAATCAGCCCTCTG	GAGGACCTGGGAGTAGATGAG
IL-1β	AGCTACGAATCTCCGACCAC	CGTTATCCCATGTGTCGAAGAA
IL-6	ACTCACCTCTTCAGAACGAATTG	CCATCTTTGGAAGGTTCAGGTTG
RPS3	GCTGAAGATGGCTACTCTGGA	ACAGCAGTCAGTTCCCGAATCC

## Data Availability

Data are contained within the article.

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
