# Peer review of "Quercetin Prevents LPS-Induced Oxidative Stress and Inflammation by Modulating NOX2/ROS/NF-kB in Lung Epithelial Cells"

_molecules, 2021, doi:10.3390/molecules26226949_

Round 1

Reviewer 1 Report

The investigation of the effects of quercetin on oxidative stress and inflammation in lung 12 epithelial A549 cells, described in the manuscript of Ok-Joo Sul and Seung-Won Ra, has been approached with the pertinent methodology and the results are clearly described.

The data demonstrate that 18 quercetin decreased ROS-induced oxidative stress and inflammation by suppressing NOX2 production and the results are of interest from a medical point of view.

However, in my opinion, to be publishable as an article, it is necessary to expand the discussion comparatively with other active pharmaceutical ingredients used for the treatment of the same diseases.

Otherwise it could be published only as a brief note or communication.

Author Response

Dear Reviewer of molecules,
“We thank you for your comments on our manuscript “Quercetin prevents LPS-induced oxidative stress and inflammation by modulating NOX2/ROS/NF-kB in lung epithelial cells.”

We revised our manuscript according to reviewers’ comments as following.

We attached the comments of the reviewer verbatim in bold with our responses in red text under each comment.

Best regards,

Ok-Joo Sul

Reviewer 2 Report

This paper shows that quercetin can prevent LPS-induced OS and inflammation in a lung cell model. Suggestions for improvement are detailed below.

1) In the abstract, it may be too much of a stretch to say that quercetin can slow the progression of lung disease.   

2) In the Introduction (or methods) the authors should refer to Figure 1A and explain the structure of quercetin – otherwise delete the figure.

3) In the description of the cell culture on lines 77-79, was the quercetin removed before treatment with LPS or left on?

4) MTT should be defined.

5) Figure 1– was there a no quercetin, LPS+ group? If not, can one be added? In other words, does LPS alone decrease viability? Also the concentrations for quercetin in the caption are incorrect, there is no 30uM and 100 should be added.

6) Although it is said that the dose of quercetin used in subsequent experiments is 10uM, there is no rationale why this dose was chosen (and not for example 20uM). Also what is the rationale for the chosen LPS dose in subsequent experiments.

7) The values in the figures are not + SD but rather + SD. They also look small for SDs, are the authors sure they are not SEM?

8) Line 148 should say Figure 1B not 2B.

9) It would be better for the figures to match in terms of LPS and quercetin. For example, Fig 2 has the LPS doses grouped, but Fig 2 has the quercetin doses grouped. Pick one way and be consistent for all figures.

10) Figure 3 – the caption says cells received NAC or DPI then quercetin, but the figure has a – (minus) under the quercetin column, which is correct? How is the data generated for the westerns – how is B-actin factored into the results? The second figure on Fig 3A has a star over LPS+, quercetin+ group. This result should be explained (and the caption showing the groups should be under the figure to be consistent with all other figures, not above it).

11) On line 175, the authors say that NOX2 elevation was almost completely reduced after quercetin treatment. Reduced to what? And is this true for the western result (there is a * above this group – see comment 10 above).

12) Throughout the paper, the authors say that NAC and DPI decreased ROS or inflammation, and so did quercetin, suggesting that they work through the same mechanisms. How does one follow the other? Couldn’t there be another mechanism through which quercetin works?

13) Figure 4 – I think the y-axis label is incorrect for the 5th figure – this an ELISA result, not PCR, correct? The caption fails to mention that NAC was given in the PCR studies shown in Fig 4A. Why are there different doses of LPS in these studies?   

14) Figure 5 – the caption should be checked for accuracy and completeness. The reader needs more information as to what is going on in this figure, either in the caption or the text of the results. Line 212, say Figures 5A and B (no comma). What is the rationale for using lamin and actin, and why are the results different? What does the 5 and 10 mean under quercetin line? What do the stars above fig 5B mean? This figure is confusing and should be discussed in more depth so the reader can follow it.

15) Line 238, the word whether should be changed to that.

16) Line 246, delete the word of.

17) The discussion is just a superficial recap of the results. It should be edited to put the study in the context of the literature.

Author Response

(The authors gave the same response as above.)

Round 2

Reviewer 1 Report

The revised version of the manuscript can be published as it is.

For the future research work of the authors to achieve the ultimate goal that "quercetin could be a potential therapeutic agent for lung diseases associated with inflammation and oxidative stress", my opinion is that more drastic concentrations of LPS would be necessary to see what is the true neuroprotective role of quercetin.

In the References section, the numbers appear twice for each of them.